# Role of Phase Information Propagation in the Realisation of Super-Resolution Based on Speckle Interferometry

Yasuhiko Arai

Department of Mechanical Engineering, Faculty of Engineering Science, Kansai University, 3-3-35, Yamate-cho, Suita, Osaka 564-8680, Japan; arai@kansai-u.ac.jp

**Abstract:** Super-resolution technology is important not only in bio-related fields but also in nanotechnology, particularly in the semiconductor industry, where fine patterning is required and super-resolution is essential. However, observing microstructures beyond the diffraction limit proposed by Abbe and Rayleigh is considered impossible because of diffraction in traditional optical microscopy observation techniques. However, in recent years, it has been possible to observe microstructures beyond the Rayleigh criterion by analysing the phase distribution of light. This study investigated the physical reasons why phase analysis makes this new analysis technique possible using simulations. The results confirmed that the phase component of the zero-order diffracted light reflected from the microstructure and able to pass through the lens system contained phase information related to the shape of the measured object. Analysis of this information demonstrates the possibility of realising super-resolution based on speckle interferometry.

**Keywords:** super-resolution; speckle interferometry; zero-order diffracted light; phase information

## 1. Introduction

Since the 2014 Nobel Prize in Chemistry was awarded to E. Betzig, W. E. Moerner, and S. W. Hell for their studies on PALM [1] and STED [2,3] using fluorescent proteins, super-resolution technology in biotechnology has become an important technology for supporting further human development [4–11]. It is also an important technology in the semiconductor industry, where much finer patterning is required in the field of nanotechnology [12,13].

However, as shown by Abbe [14] and Rayleigh [15,16], it has been proposed that image acquisition with TV cameras, which can capture 2D information instantaneously, has limited use for the observation of microstructures due to the diffraction limit as a lens property. According to the Rayleigh criterion based on this idea, structures finer than approximately half the wavelength of the light source used for observation cannot be observed using a lens system.

However, a new super-resolution technique based on speckle interferometry that analyses the phase distribution of light was recently proposed by Arai [17]. This technology is based on speckle interferometry, which analyses the phase distribution of light to measure deformation with a resolution better than one hundredth of the source wavelength. It is therefore fundamentally different from the conventional technique proposed by Abbe and Rayleigh, which is based on the analysis of the intensity distribution of light. It also differs from the super-resolution technique using fluorescent proteins. The principle of this method is described in Chapter 2.

It is now known that phase analysis techniques using a laser beam can easily detect the phase distribution of light with high resolution, including fringe-scanning techniques using light interferometry [18,19]. It is also well known that speckle interferometry is an excellent measurement technique, especially when the measurement object (for example: a diffraction grating) is a structure with a rough surface, where high measurement resolution can be expected [20].

However, even if the phase resolution of the speckle interferometry is high, the three-dimensional shape cannot be observed unless the shape of the measured object is captured with the detector as information on the diffraction phenomenon. In the Arai method, the light from the measured object is only zeroth-order diffraction light, and if there is no shape information in the zeroth-order diffraction light, the high phase resolution of the speckle interferometry method cannot be used.

In this paper, based on the idea that phase information may be preserved in this zeroth-order diffracted light, the physical properties of the phase distribution of the light to achieve super-resolution in the Arai method were investigated using electromagnetic field simulations. Consequently, the mechanism for achieving super-resolution was physically clarified.

## 2. Materials and Methods

### 2.1. Principle of Measuring Microstructures beyond the Diffraction Limit Based on Speckle Interferometry Technology

Assuming, for example, that the measurement cross-section of the measurement object shown in Figure 1a can be expressed as f(x) and that a lateral shift δx is assigned to the measurement object, as shown in Figure 1b, the shape displacement occurring at each measurement point can be expressed as f(x + δx) − f(x). The displacement of the shape at each of these points is then accurately measured using speckle interferometry [20–22], and the pseudo-differential value {f(x) − f(x + δx)}/δx in the shift direction with respect to the shape is obtained by dividing the detected displacement by the transverse shift amount. Furthermore, the shape of the measurement object can be reconstructed by integrating the pseudo-differential value [17,22].

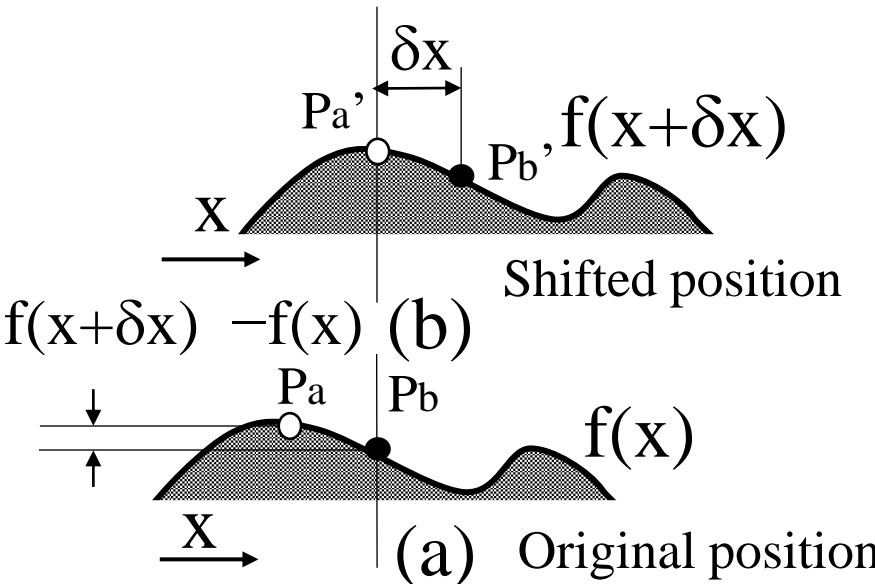

**Figure 1.** Principle of analysing method. (**a**) Section of measured object at original position. (**b**) Section of measured object at shifted position.

Through this computational process, structures beyond the diffraction limit were observed. Speckle interferometry is a method for detecting the amount of deformation with high resolution by capturing image information before and after the deformation of the measurement object [18,20]. Therefore, speckle interferometry was a suitable analysis method for detecting the shape displacement f(x + δx) − f(x) at each measurement point on the measuring object because of the lateral shift of the measuring object at a high resolution in this study. This property was applied to the super-resolution technique in this study [17]. However, the actual calculation process uses an algorithm that does not actually require a lateral shift operation [21], as the images acquired in the computer are virtually shifted

horizontally in memory. In this way, it is possible to observe microstructures with only a single speckle pattern. Dynamically moving objects can be observed using this technique.

### 2.2. Observation Optics

The basic optical system used in this experiment, which is based on speckle interferometry, is shown in Figure 2a. The speckle pattern taken with this optical system is shown in Figure 2b. This optical system uses a laser with a wavelength of 532 nm and an output of 100 mW as the light source and a Mitsutoyo (M Plan Apo 200× manufactured) objective lens.

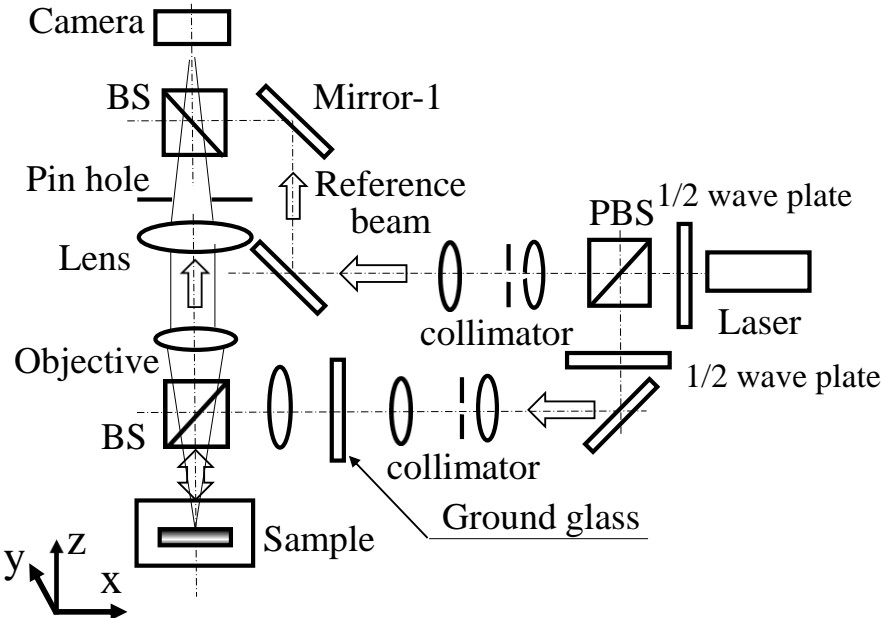

**Figure 2.** Experimental optical system. (**a**) Schematic of experimental optics. (**b**) Speckle pattern.

This lens has a magnification of 200× and a numerical aperture of 0.62, which provides a diffraction limit of 523.4 nm (=0.61 × 532 nm/0.62). The camera has an image element spacing of 1.6 μm and a pixel size of 1024 × 1024, and 4096 grey levels [17]. Furthermore, the use of plane waves as the reference light of the speckle interferometer enables deformation measurements to be performed using only two speckle patterns before and after deformation. This has been demonstrated in previous reports [17,21–23].

### 2.3. Simulation of the Analysis in the Observation of Microstructures Based on Phase Analysis
2.3.1. Simulation Model

In general, it is extremely difficult to completely eliminate disturbances such as stray light in an optical system when using an actual optical system to verify and confirm the principle, as attempted in this study. In addition, it is difficult to discuss the physical phenomena in detail because of limitations such as measurement accuracy and the experimental environment.

To avoid these problems, in the current study, I used an electromagnetic field simulation software (COMSOL Multiphysics) to investigate the physical phenomena in which the observation of microstructures beyond the diffraction limit was achieved using a new super-resolution technology based on speckle interferometry techniques [21–23]. A computer simulation model used in a previous report [23], shown in Figure 3a, was used as the optical system model in this study.

Employing the optical system used for the experiment shown in Figure 2, an experiment was performed by analysing the light reflected from a glass reflective diffraction grating as a measurement sample [17]. Because this study investigated the physical interpretation of the properties of the phase distribution of light from microstructures beyond the

diffraction limit in super-resolution speckle interferometry techniques, the measurement object was used as a simplified model compared with the treatment in previous reports [23].

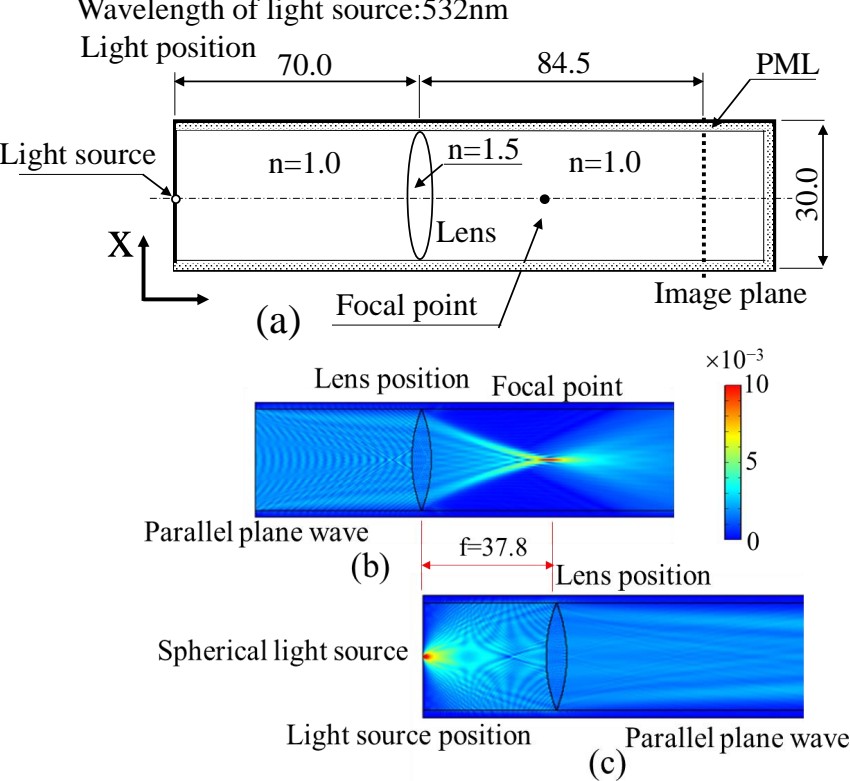

**Figure 3.** Simulation model of the optical system shown in Figure 2. (**a**) Optical system in simulation approach. (**b**) Confirmation via electromagnetic field strength distribution of the focal state of light from a plane-wave light source after passing through the lens. (**c**) Confirmation via electromagnetic field strength distribution of the parallel state of light from a point light source placed at the front focus of the lens after passing through the lens.

Specifically, the shape was set as a simple sample modelled by supplying a group of spherical-wave point sources with a phase distribution depending on the shape given to the reflected light from each point on the measurement object. In other words, when the measurement object is a flat plane, as shown in Figure 3b, it is modelled by assuming that spherical-wave light sources with the same phase are aligned on the same plane. When a surface with protrusions/steps is to be measured, a spherical-wave light source with the wavelength of the light source set to $2\pi$ rad is used to correspond to the height of the protrusions, and the phase given according to the shape is attributed to the protrusions, thus expressing the shape of the protrusions as a phase distribution.

In the simulation model shown in Figure 3a, the mesh size was set by considering the load on the computer memory, as in a previous report [23]. In other words, 1/12 of the wavelength was set as the mesh size as a condition under which the influence of the mesh size did not extend to the calculation results as far as possible. Under these conditions, it was confirmed that the calculation results did not change even if the mesh was not engraved any finer. The arrangement of the light sources as measurement objects was also set with a minimum unit spacing of 0.1 nm, considering the load on the computer's memory. To make effective use of the limited memory, the parallel side walls and the right-side wall of the computational domain were defined as perfectly matched layers (PMLs). PLMs can be used to remove the effect of reflected waves from the walls and improve the accuracy of the analysis when performing calculations in a limited computational domain. In creating the simulation model, as in a previous report [23], the computational domain was defined

with the minimum possible memory (memory capacity of 2 TB) to minimise the load on memory capacity.

The light source used in the simulation was a spherical-wave source derived from the Maxwell equations defined by Equation (1) [23]. In the light source model representing the electromagnetic field strength E shown in Equation (1), the unit of amplitude $Va$ is V/m (electromagnetic field strength), $\lambda$ is the wavelength (:nm), and $\phi$ is the initial phase (rad) of the light from the source. The distribution based on this phase $\phi$ is used to set the shape of the measurement object as described above. $E$ is electromagnetic field strength.

$$E = V_a \times \frac{\exp\left\{i\left(\frac{2\pi}{\lambda}\sqrt{x^2+y^2}+\phi\right)\right\}}{\sqrt{x^2+y^2}} \tag{1}$$

The specific optical elements used in the simulation model were modelled assuming the objective lens was a thin biconvex lens and defining the refractive indices of air and the lens as 1.0 and 1.5, respectively. First, the focal lengths and diffraction limits of the lenses of the optical system were identified.

The focal length (f = 37.8 µm) was determined by setting the lens focal point as the point where the highest intensity of the electromagnetic field is focused by the lens when the plane wave as a collimated light is irradiated by the lens from the left wall surface, as shown in Figure 3b. Furthermore, the focal length f was confirmed as 37.8 µm by setting the spherical-wave source on the optical axis of the left wall surface, as shown in Figure 3c, and confirming that the electromagnetic field strength after passing through the lens is parallel light when the lens is set at a distance from the left wall surface corresponding to the lens focal length determined in Figure 3a.

The lens used here was designed to be glass with a refractive index of 1.5 and a circular arc of radius of 40 µm. As both convex surfaces of the lens are formed by arcs of radius of 40 µm, the focal length can be determined as 40 µm if the thickness of the lens is sufficiently thin [23].

However, in this study, the focal length was determined using a procedure that considers the relationship between the actual collimated light and the lens, as shown in Figure 3, because the thickness of the lens was not necessarily sufficiently thin (5.83 µm relative to the lens diameter). Consequently, the NA of the objective lens was estimated as 0.37 [=1 × sin (tan$^{-1}$ (15/37.8))]. The diffraction limit as a Rayleigh criterion could then be obtained as 877 nm (=0.61 × $\lambda$/NA = 0.61 × 532/0.37). The characteristics of this optical system were based on the results discussed in a previous report [23] and were also used in this study.

### 2.3.2. Verification of the Rayleigh Criterion in the Diffraction Limit

In the optical system shown in Figure 3, when the spherical-wave light sources shown in Equation (1) are placed at symmetrical positions across the optical axis on a plane 70 µm away from the lens, separated from each other by 2 µm, and the phase ϕ of both light sources is 0 rad, the light from the two light sources shown in Figure 4a interferes and forms Young fringes because it does not exceed the diffraction limit.

A part of Young's fringes passes through the lens, as shown in Figure 5a, which shows the two-dimensional distribution of the electromagnetic field strength, and bright spots can be observed as two points with two intensity peaks on the imaging plane, as shown in Figure 4b.

Conversely, if the phase is 0 rad when the distance between the two light sources shown in Figure 4c is 0.2 µm, the light from the two light sources does not reach the lens, except for the zeroth-order diffracted light along the optical axis, as shown in Figure 6a, which shows the two-dimensional distribution of the electromagnetic field strength.

In this case, only the zeroth-order diffracted light along the optical axis reaches the lens because the two light sources are beyond the diffraction limit.

Consequently, as shown in Figure 4d, the intensity distribution on the image plane becomes a single point, which means that, in this case, it cannot be observed as two bright points.

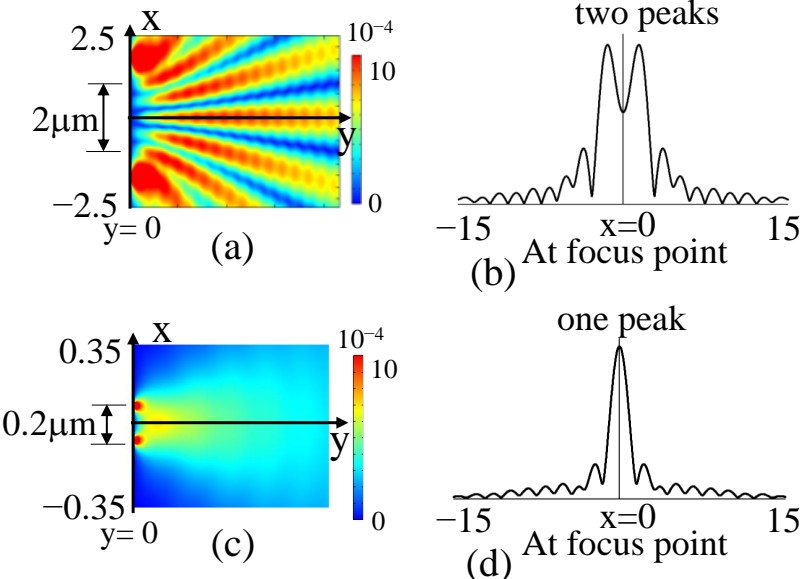

**Figure 4.** Relationship between distance of two light sources and diffraction phenomena at the diffraction limit of 877 nm. (**a**) When the distance (2 μm) between the two light sources is larger than the diffraction limit (877 nm), Young's fringes are observed. (**b**) Two intensity peaks can be observed at the imaging plane when the distance (2 μm) between the two light sources does not exceed the diffraction limit (877 nm). (**c**) When the distance between the two light sources (0.2 μm) is less than the diffraction limit (877 nm), Young's fringes are not observed. (**d**) Only one intensity peak can be observed at the imaging plane when the distance (0.2 μm) between the two light sources exceeds the diffraction limit (877 nm).

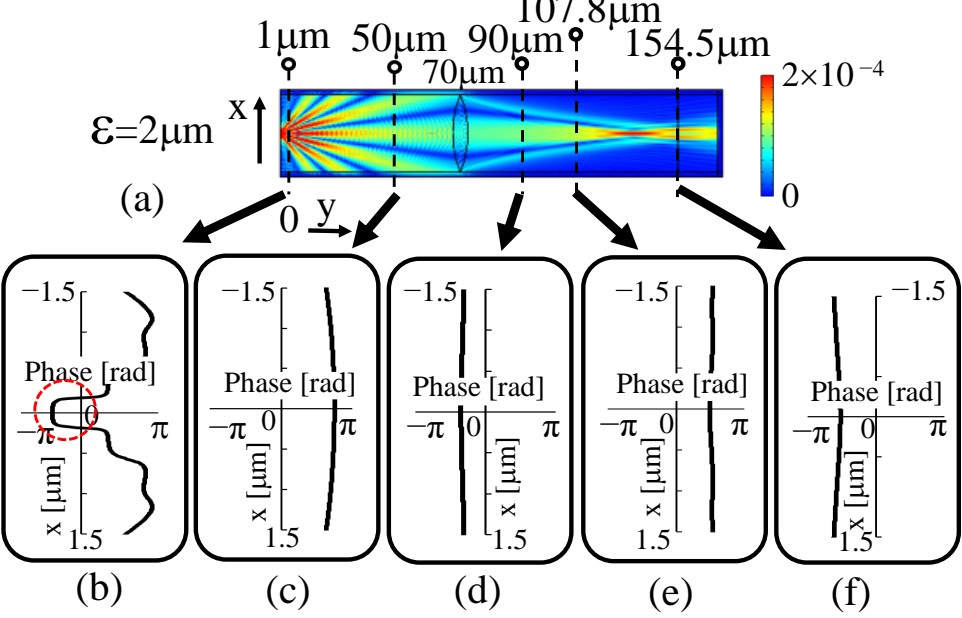

**Figure 5.** Phase distribution at each measurement point ($\varepsilon$ = 2 μm). (**a**) Distribution of electromagnetic field strength. (**b**) Phase distribution at 1 μm. (**c**) Phase distribution at 50 μm. (**d**) Phase distribution at 90 μm. (**e**) Phase distribution at 107.8 μm (focal point of this lens). (**f**) Phase distribution at 154.5 μm (image plane of this lens).

Thus, it can be confirmed that the diffraction-limited phenomenon, in which the observation of two points becomes impossible depending on the wavelength and distance of the light source, can also be realised in the simulation, as shown in Figure 4.

In this case, as shown in a previous report [23], even if the distance between the two light sources narrowly exceeds the diffraction limit, if the phases of the two light sources are shifted by $\pi$ rad, they can be observed as two bright points instead of as a single point as the Rayleigh criterion suggests. Furthermore, this phenomenon has been confirmed by other institutes [24–26].

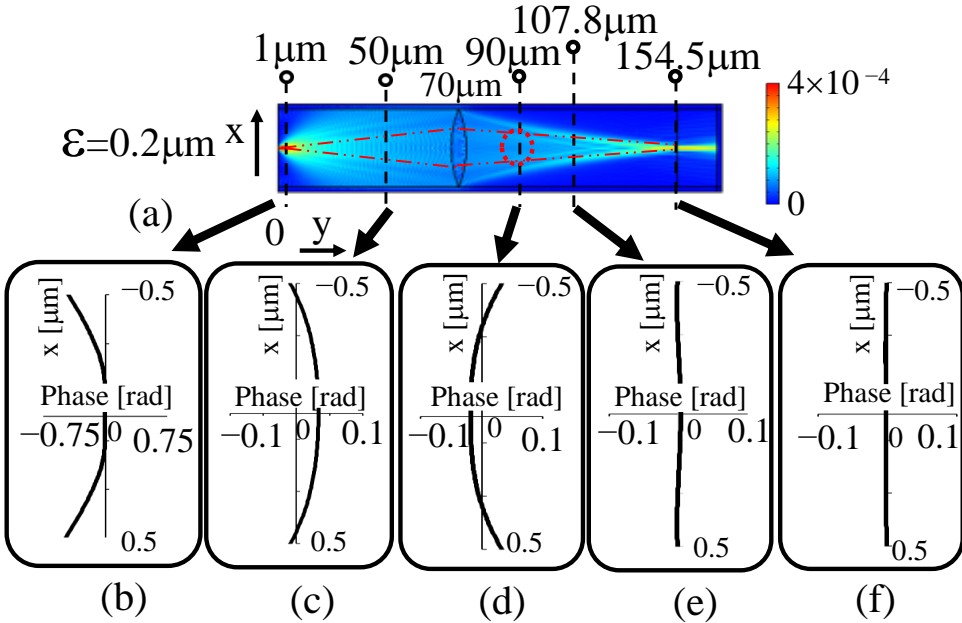

**Figure 6.** Phase distribution at each measurement point ($\varepsilon$ = 0.2 μm). (**a**) Distribution of electromagnetic field strength. (**b**) Phase distribution at 1 μm. (**c**) Phase distribution at 50 μm. (**d**) Phase distribution at 90 μm. (**e**) Phase distribution at 107.8 μm (focal point of this lens). (**f**) Phase distribution at 154.5 μm (image plane of this lens).

## 3. Results and Discussion

### 3.1. Observation Results for Diffraction Gratings beyond the Diffraction Limit

The results of the observation of a reflective diffraction grating with a periodic structure of 278 nm, in which the period observed using SEM exceeded the diffraction limit, are shown in Figure 7a. Figure 7b shows the results of measuring this reflection-type diffraction grating using the optical system shown in Figure 2a. Furthermore, this method can be used not only to observe the microstructure of the repeating structure shown in Figure 7 but also to capture three-dimensional structures, such as microspheres and microcharacters [22], as reported previously.

In contrast, the results shown in Figure 7b indicate a slightly magnified period compared with the scanning electron microscope (SEM) observation results. Because this study was not based on traditional observation techniques and the object was not observed by focusing the object on the imaging plane, the phenomenon of this magnification change could not initially be interpreted clearly with regard to the observation magnification of the lens. However, the current simulation also examined the phenomenon occurring with a change in the magnification.

### 3.2. Phase Distribution at Each Point on the Optical Axis in the Case of Exceeding the Diffraction Limit in Simulation

The state of the phase distribution of the light from the microstructure beyond the diffraction limit at each point on the optical axis in relation to the two light sources and their distances was then verified using a simulation.

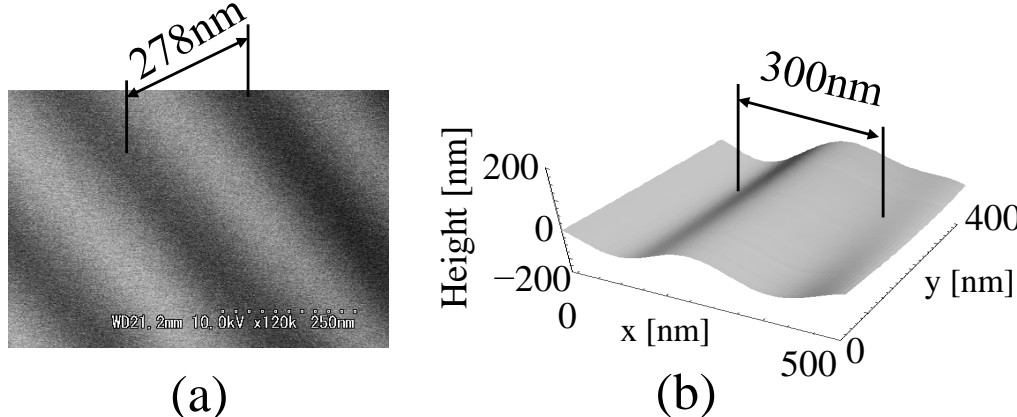

**Figure 7.** Measured object and results using this optical system. (**a**) SEM (×120,000) image of diffraction grating (period: 278 nm). (**b**) Results obtained using optical system shown in Figure 2.

Figure 5a illustrates the two-dimensional distribution of the electromagnetic field strength near the optical axis for two light sources with a distance of 2 μm between two points that did not exceed the diffraction limit when both phases were 0 rad. At 1 μm, immediately after being emitted from the light source, the lights interfered, as shown by the red dashed line in Figure 5b, which shows the phase distribution in the x-direction. It can be noted that the light is a plane wave with a flat phase depending on the width of the beam along the optical axis.

As shown in Figure 5a, this interrupted zero-order diffracted light travelled towards the lens with a slight spread along the optical axis. Consequently, as shown in Figure 5c, at 50 μm from the measurement object, the phase distribution of the zero-order diffracted light grew into a spherical wave whose phase was slightly delayed as it travels away from the optical axis because the light source itself was set as a spherical wave.

After passing through the lens at 90 μm, as shown in Figure 5d, the direction of convexity of the phase distribution changed so that the phase distribution at 50 μm was reversed by the lens. The phase advanced at the periphery, away from the optical axis. It is shown that the light itself became a wavefront focused towards the focal point.

Then, at 107.8 nm in the back focus of the lens, the phase was almost flat near the optical axis, as shown in Figure 5e. Furthermore, on the image plane in Figure 5f, the phase distribution can be seen to be slightly spherical and convex in the direction of travel.

However, when the phases of the two sources shown in Figure 6a were both 0 rad and their distance was 0.2 μm, the phase distribution did not become like the zero-order diffracted light of the Young's fringe as it does when the diffraction limit seen in Figure 5a was not exceeded. In this case, the phase was delayed in the periphery with a symmetrical phase distribution across the optical axis in Figure 6b, which shows the state at 1 μm immediately after the light source was activated. It can be observed that the light wave was largely spread out.

However, in Figure 6, Young's interference fringes cannot be observed, as also shown in Figure 5. Therefore, light from the two light sources diffused immediately in the transverse and travelling direction near the light source. As a result, the change in the phase distribution of light in the direction of travel was small, and the phase distribution was average.

For this reason, the maximum and minimum values of the phase distribution in the direction of travelling were ±0.1 rad, much smaller than the values in Figure 6, as compared with the values of π rad and −π rad shown in Figure 5. These indicate that the state of the spread of the phase distribution is clearly different when the distance between the light sources exceeds the diffraction limit and when it does not.

At the 50 μm position shown in Figure 6c, where the light progressed farther from the position shown in Figure 6b, the phase was delayed at the position away from the optical

axis and the range of the peripheral phase delay was expanded as a spherical wave with a broad spread. At the 90 μm position shown in Figure 6d, after it had passed through the lens, contrary to Figure 6c, the phase at the periphery away from the optical axis increased and the phase distribution became a wavefront (phase distribution) that focused towards the rear focal point. At the back focus of the lens, as shown in Figure 6e, the phase was a flat plane wave on the optical axis. On the imaging plane, as shown in Figure 6f, it can be noted that the phase was again a plane wave with a flatter phase. This state indicates that when lights from two light sources exceeding the diffraction limit are captured on the image plane, as shown in a previous study [23], a phenomenon occurs that is in agreement with the results obtained when the phase is spatially flattened on the image plane, even if the phases of the two light sources are different.

Thus, if the phase is spatially flattened on the image plane, it can naturally be understood that the phase difference between the two points to be measured cannot be detected by observing the phase distribution on the image plane. However, it was confirmed that in actual optical systems, as shown in Figure 2, when a three-dimensional shape exists on the measurement object, the phase distribution can be detected, and the shape of the object can be reconstructed as a phase distribution, as shown in Figure 7.

It should be noted that the experiments did not use spherical waves with phase-aligned wavefronts, as in the simulations. In other words, scattered light with wavefronts of random phases was used in the experiment. In addition, it should be noted that the phase distribution was not detected at the image formation position shown in Figure 6a in the simulation.

However, in the current experiment, the phase information was detected at a position that was not too close to the objective lens. Based on the experimental situation, it was also decided the phase in the simulation would be detected at a position not too close to the objective lens and at a position where the phase changed significantly in the *x*-axis direction with respect to the direction of light travel. As a position suitable for this condition, the phase was detected at 90 μm, as shown in Figure 6d, which is considered to be the optimal position for detecting the phase information on a measured object as it is the least affected by the surroundings.

### 3.3. Phase Analysis in the Area between the Lens Position and the Posterior Focal Point

The distribution of the electromagnetic strength in Figure 6a shows that when the distance between the two light sources was 0.2 μm, which is beyond the diffraction limit, the light from the light source passed through the lens while only the zeroth-order diffracted light spread slightly radially from the light source. In other words, light from the light source spread radially as a spherical wave immediately after it was emitted, as shown in Figure 6b,c. To obtain information on a spread of light using its intensity distribution, the light was focused again using a lens to form an image of the object, which was then observed as an image. However, in the analysis using the phase distribution, when the spreading light was refocused, the phase changed according to the length of the optical path through which the light had passed and was superimposed on the phase distribution, containing information concerning the shape of the original measurement object in the vicinity of the optical axis. Consequently, it may be difficult to detect the phase accurately based on the original shape of the measurement object.

It is thought that the effect of the optical path length of the spreading light may cause the original information to be hidden in the phase distribution due to the new optical path length.

Therefore, it is essential to capture only the light near the optical axis of the zeroth-order diffracted light, which has as little spread as possible and is not focused by the lens in cases such as those shown in Figure 6a. In actual optical systems, the principle of measurement is based on speckle interferometry. Therefore, it is believed that the light incident on the lens is only detected near the optical axis passing through the pinhole. In other words, if only light on an optical axis parallel to the optical axis is detected as

far as possible, as shown in Figure 6a, such that no additional phase element owing to differences in the optical path is added to the phase distribution of the measurement object, the observation of the phase distribution may be close to experimental results obtained using actual optical systems. However, it was assumed that the COMSOL software used in this simulation would facilitate a pinhole setting by placing a wall with a PML setting just in front of the lens. However, the light reflected from the pinhole wall caused multiple complex reflections in the optical system because the PML settings were not sufficiently set. In other words, the problem is that the pinhole setting is not always perfect. Therefore, it was decided that a simpler process would be performed instead of a pinhole-based simulation, by capturing only the light on the optical axis as far as the area indicated by the red dashed circle in Figure 6a at 90 μm, as shown in Figure 6d, as pinhole-based simulation is currently difficult to perform.

Under the conditions of this information collection, as shown in Figure 8, two light sources (Figure 8a) and three light sources (Figure 8b) were assumed in this study. In this case, the distance between the light sources was set to ε μm, and the phase of each light source was set to α, β. Furthermore, using the knowledge that 2π rad corresponds to the light source wavelength of 532 nm, the position of the light sources was set approximately by changing the value of the phase. Specifically, it was decided that the light would come from a source approximately one half of a wavelength in front (266 nm) of the phase when it advanced by π rad. The spherical wave from the light source set up in this way passed through the lens, and the phase was observed at a distance of 90 μm (furthermore, the phase difference was converted into a length using the wavelength–phase relationship).

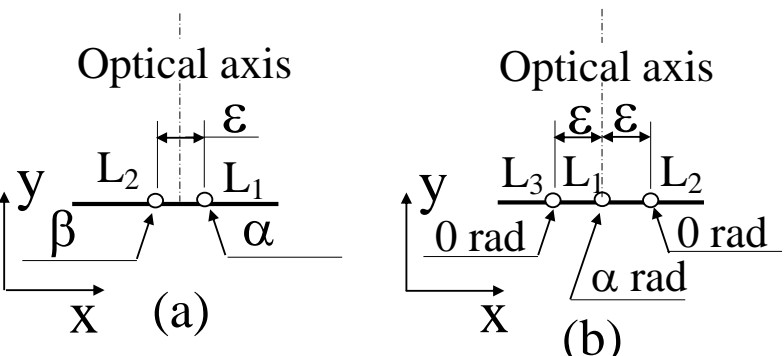

**Figure 8.** Position of light sources in simulation. (**a**) In the case of two light sources. (**b**) In the case of three light sources.

In the optical system used in the experiments, the phase distribution of the object light was detected with high resolution based on the spatial fringe analysis method by interfering with the object light from the object and using a plane wave as the reference light [17]. Based on this approach, in the simulation, the wavefront corresponding to the reference light was calculated in advance as the phase distribution when both phases, α and β, of each light source in Figure 8 were set to zero. By subtracting such a reference phase distribution from the phase distribution obtained by setting α and β, only the phase distribution originally intended to be obtained was calculated.

### 3.4. Simulation of Two Light Sources with Different Phases

Either phase α or β of the two light sources separated by ε = 0.2 μm shown in Figure 8a was set to 0 rad, while the other was set in the range of angles π/4, π/2, and 3π/4 in the first and second quadrants, where the phase change in the phase plane was relatively small.

As a result of using this condition, information on the simpler nature of the state, which was close to linearity, eliminating the complex nature; the nature of the phase distribution, was considered to be beyond the diffraction limit, which was the object of study of this research, can be successfully verified by observing it without complicating it. As a concrete matter of consideration, the phase distribution of the wavefront detected at 90 μm when

one of the values of α and β was set to 0 rad and the other was changed to π/4, π/2, or 3π/4 rad was examined.

Figure 9a shows the phase distribution when both α and β were set to 0 rad. This condition was used as the standard phase distribution for the reference beam. In the experiments, speckle interferometry was used to determine the displacement, f(x + δx) − f(x), at each measurement point when the object was laterally shifted by performing a calculation between the two measurement points of the object. Based on the above concept, to determine the amount of change, φαβ, when either α or β was changed to 0 and the other to π/4, π/2, or 3π/4 rad, the phase distribution, φ00, when both α and β were set to 0 rad was obtained in advance. The value of φ00 was then subtracted from the value obtained when one of them was changed to π/4, π/2, or 3π/4 rad, respectively. This process can be used to accurately detect the phase change, φαβ, associated with a change in α and β.

Furthermore, the phase distribution in Figure 9a is shown to be a convex downwards curve. This phenomenon can be confirmed by observing the electromagnetic field distribution at a distance of 90 μm from the measurement object in Figure 6a, where the light is focused towards the optical axis by the lens. The phase distribution is considered to be a convex downwards curve because the phase is farther away from the optical axis, owing to the different optical path lengths between the light from the periphery of the lens, which is farther away from the optical axis, and the light near the optical axis. This result was also observed in the phase distribution shown in Figure 6d. In other words, phase information based on the shape of the measurement object cannot be properly extracted without cancelling out the optical path length changes caused by lens focusing. Such a process has been realised in experiments using speckle interferometry.

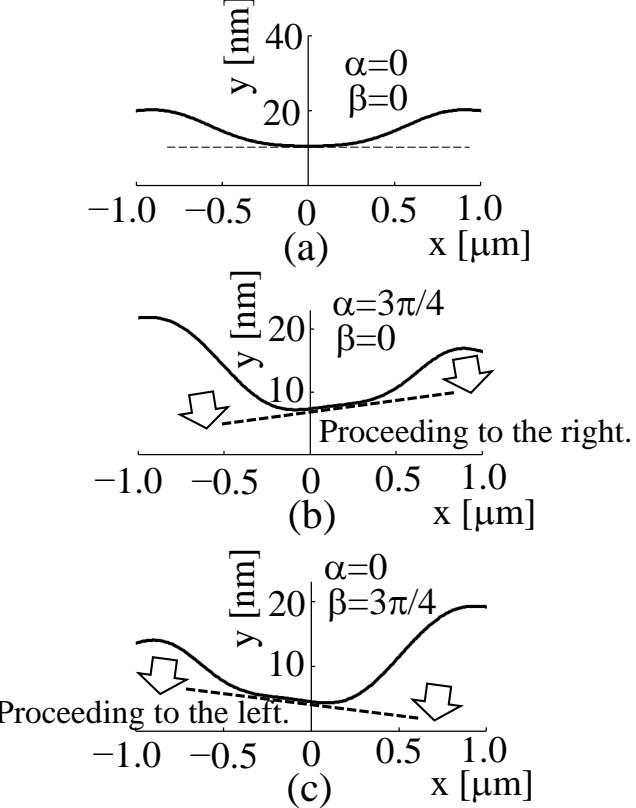

**Figure 9.** Phase distribution in the case of two light sources at 90 μm. (**a**) Both phase values of α and β are zero rad. (**b**) α = 3π/4 and β = 0. (**c**) α = 0 and β = 3π/4.

Figure 9b shows the phase distribution when $\alpha = 3\pi/4$ and $\beta = 0$. Since $2\pi$ rad corresponds to a wavelength of 532 nm, the phase distribution was converted to a length on the *y*-axis.

In this phase distribution of the object light, the angles of $\alpha$ and $\beta$ were already different, so the phase distribution converted into the length of the measurement result was asymmetrical on the left and right. The distortion of the phase distribution in this oblique direction confirmed that the phases of the two light sources under examination were different. Furthermore, Figure 9c shows the result for $\alpha = 0$ and $\beta = 3\pi/4$. In this case, the angles of $\alpha$ and $\beta$ were opposite to those shown in Figure 9b, indicating that an inverse asymmetric phase distribution was obtained.

Figure 10 shows the results of replacing the phase difference with the length when the phase distribution was subtracted, as shown in Figure 9a, with both $\alpha$ and $\beta$ set to 0, as seen in Figure 9b,c, with different angles of $\alpha$ and $\beta$.

The different phases of $\alpha$ and $\beta$ on the left and right in Figure 10a,b, 0 and $3\pi/4$, show that the slope of the detected phase distribution changed from 2.96 (nm/μm), as shown in Figure 10a, when the wavefront travelled in the right direction, to $-3.13$ (nm/μm), as shown in Figure 10b, when the wavefront travelled in the left direction. The absolute value of the slope changed slightly, but the sign also changed. The difference between the phases of $\alpha$ and $\beta$ could be confirmed by checking the phase distribution at 90 μm.

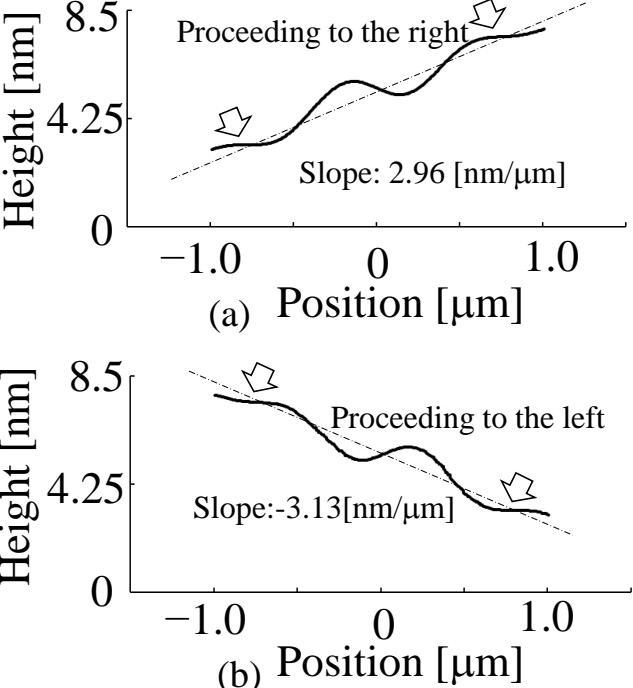

**Figure 10.** Investigation of the direction of propagation of wavefronts in the case of two light sources with different phases. (**a**) $\alpha = 3\pi/4$ and $\beta = 0$, as in the case shown in Figure 9b. (**b**) $\alpha = 0$ and $\beta = 3\pi/4$, as in the case shown in Figure 9c.

Figure 11 shows the change in the slope of the phase distribution as shown in Figure 10 when the values of phase $\alpha$ and $\beta$ of the two light sources in Figure 8 were varied as $\pi/4$, $\pi/2$, and $3\pi/4$ and set according to the size of the circles and the colour (black or white), respectively. Then, the case where values of 0.05, 0.1, 0.2, and 0.5 μm were set for the horizontal axis of the distance between light sources was investigated. The phase of the two light sources changed, generating a change in the slope of the wavefront in accordance with the phase difference between the two light sources. Furthermore, the tilt of the wavefront increased with increasing distance between the sources, as shown on the horizontal axis.

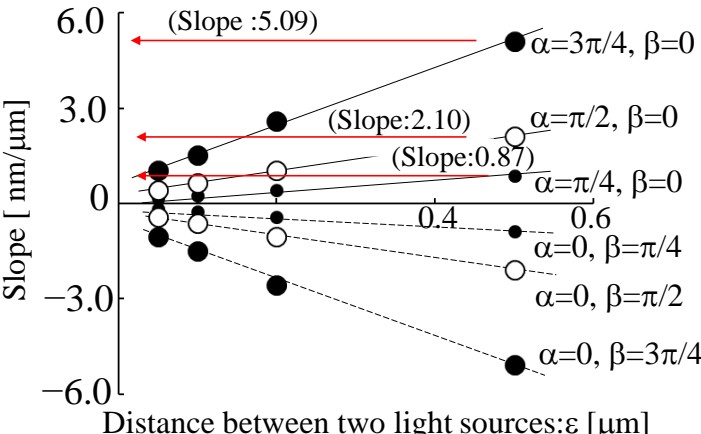

**Figure 11.** Relationship between slope of wavefront of light and distance of light sources, with the phase value at each light source shown.

In other words, the results in Figure 11 show that the slope of the wavefront varies with changes in the phase of the light source and with the distance between the light sources, which is related to the position of the light source. In this case, according to the calculations presented in this study, because the changes were limited to a minute range, it was found that there was a proportional relationship between the intensity of the light sources and the slope of the wavefront. Furthermore, the changes in the phase of the light sources and the slope of the wavefront shown by the large●, large○, and small● were slope = 5.09 nm/μm for the large●, slope = 2.10 nm/μm for the large○, and finally slope = 0.87 nm/μm for the small● shown in brackets for $\varepsilon$ = 0.5 μm when $\beta$ = 0. It can be seen that the slope was approximately related to the magnitude of the phase $\alpha$ of the light source ($\pi/4$, $\pi/2$, $3\pi/4$).

As shown in Figure 8a, it can be assumed that the phase distribution of light from the two light sources immediately after passing through the lens contains information regarding the shape of each reflection point on the object. Based on this property, the shape of the microstructure can be detected in the experiment. In other words, it can be confirmed that there is a phase distribution in the zero-order diffracted light as light reflected from microstructures.

*3.5. Simulation of Different Phases from Three Light Sources*

In the previous section, it was confirmed that the zero-order diffracted light from two light sources with different phases in close proximity beyond the diffraction limit contained information about the phases of the two light sources.

Therefore, different phases between the three light sources in close proximity beyond the diffraction limit were further investigated.

In other words, the case where the phase of the light source L1 in the middle was varied on the optical axis, as shown in Figure 8b, with the phase of L2 and L3, which were separated by $\varepsilon$, set as 0 rad was considered. In the case of the two light sources considered in the previous section, when there was a difference in phases between the light sources, it was confirmed that the phase distribution was included in the zeroth-order diffracted light after passing through the lens because of the change in the travel direction of the wavefront. It was investigated whether the phase difference could be detected among the three light sources.

Figure 12a–c show the phase distribution at 90 μm when the respective distance between the three light sources was set as $\varepsilon$ = 0.2 μm when the phase of L2 and L3 was set as 0 rad and the phase of L1 was set as a negative delayed phase of $-\pi/8$, $-\pi/4$, and $-\pi/2$ rad. The phase of L1 was delayed in comparison with that of L2 and L3, which confirms that the phase distribution near the optical axis was delayed at 90 μm, resulting in a convex downwards result.

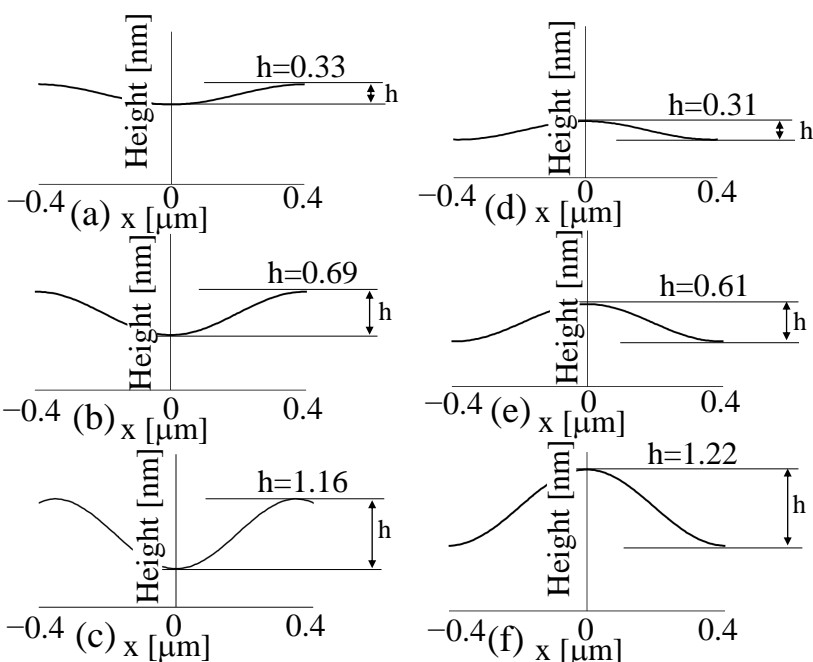

**Figure 12.** Phase distribution at 90 μm in the case of three light sources, where the midpoint light source (L1) has only one phase. (**a**) ε = 0.2 μm, α = -π/8 rad, (**b**) ε = 0.2 μm, α = −π/4 rad, (**c**) ε = 0.2 μm, α = −π/2 rad, (**d**) ε = 0.2 μm, α = π/8 rad, (**e**) ε = 0.2 μm, α = π/4 rad, (**f**) ε = 0.2μm, α = π/2 rad.

Phase distribution at 90 μm in the case of three light sources, where the midpoint light source (L1) has only one phase.

In Figure 12, 2π rad corresponds to a wavelength of 532 nm, and the phase difference between the line connecting L2–L3 and L1 is defined as h$_{diff}$, which was converted to a length, h. In this calculation, the results were acquired using the phase distribution obtained by setting the phases of all light sources, L1, L2, and L3, to zero as a standard phase distribution.

First, phases L1, L2, and L3 were set to zero, as shown in Figure 10, and then subtracted from the phase distribution when L1 was changed. On the other hand, Figure 12d–f show the results of the phase distribution at 90 μm, replaced by the length h, when L1 was set to be the phase advanced by setting the phase α to have positive values of π/8, π/4, and π/2 rad compared with L2 and L3.

In this case, in contrast to Figure 12a–c, the phase distribution in the vicinity of the optical axis advanced with respect to the surroundings.

The length h was positive, yielding an upward convex result. In this case, the process was also performed by subtracting the reference phase distribution obtained by defining the phase of all light sources as zero. It can be confirmed that the zero-order diffracted light contains phase information of the measurement object with regard to the phases of the three light sources.

The detection of phase components in zero-order diffracted light was considered to realise super-resolution technology using speckle interferometry.

Figure 13 shows the results of the phase advance and delay states in the near optical axis in the phase difference between L1 and L2, with L3 shown on the horizontal axis, converted to the length h, with the parameters of the distance between the light sources ε set to 0.1, 0.2, and 0.5 μm, as shown in Figure 12. As the values obtained in the simulations were very small, the characteristics were compared by expressing the length values as a function of the phase change using logarithms. It was found that the phase change increased as the distance ε between the light sources increased from 0.1 to 0.2 and 0.5 μm. It can also be shown that the detected length h increased as the phase difference increased

from $\pi/8$, $\pi/4$, and $\pi/2$ rad. Furthermore, when L1, as indicated by the white circle, was in a positive phase with respect to L2 and L3 (with the shape of the measured object protruding), it was slightly larger than in the case where the phase was lagging, as indicated by the black circle (the shape of the measured object was depressed).

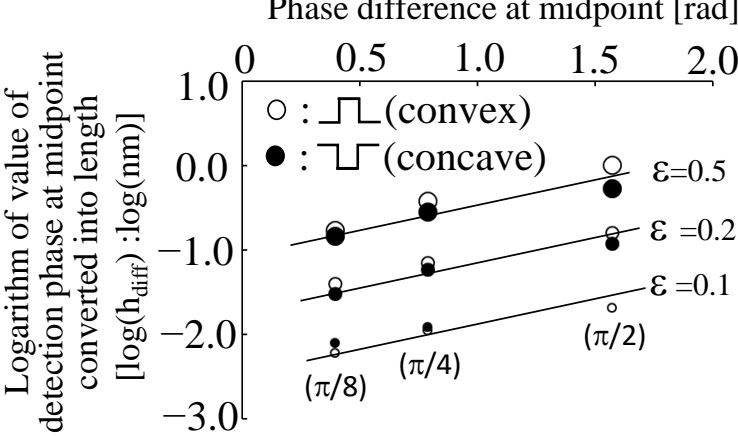

**Figure 13.** Relationship between change in phase of detected midpoint and distance of light sources, with phase values shown. (The phase was converted to length using the relationship between $2\pi$ rad and the wavelength (532 nm).).

This indicates that the protruding shape of the measured object is easier to detect. However, the results shown in Figures 12 and 13 are the results of calculations based on simulations, which resulted in the detection of lengths below the nm level.

In particular, in the case of $\varepsilon = 0.1$ μm, where the light sources were close together, the change in length h as the difference between L1 and L2–L3 was extremely small. As a result of the calculation, the zero-order diffracted light was considered to contain phase information; however, the calculated value was very small.

It is thought that this problem should be considered by setting up a pinhole in front of the lens in the simulation to detect only zero-order diffracted light with no spread near the optical axis and by simulating scattered light with a larger number of ray vectors.

In this study, to understand the influence of parameters such as light source spacing on the results, the results for three light sources using a logarithmic function were obtained, which provide a relatively good examination of the phenomenon, rather than directly comparing small arithmetic and large results.

Because it is currently difficult to set up pinholes in optical systems using COMSOL, in the future, consideration should be given to making COMSOL more similar to actual optical systems by discussing the creation of pinholes that can prevent complex reflections and interference from occurring in the optical system with software development engineers.

### 3.6. Observation of an Object with Periodic Structure

By investigating two and three light sources, it was shown that phase information is contained in zero-order diffracted light. Therefore, a measured object with periodicity, as shown in Figure 7, was considered next.

That is, a simulation was performed with an object consisting of 21 light sources with a period of 0.4 μm. The phase of each source changed in the range of $-\pi/2$ to $\pi/2$, being continuously and closely adjacent at 0.05 μm beyond the diffraction limit, as shown by the red dashed line in Figure 14. A phase distribution at a position of 90 μm was detected.

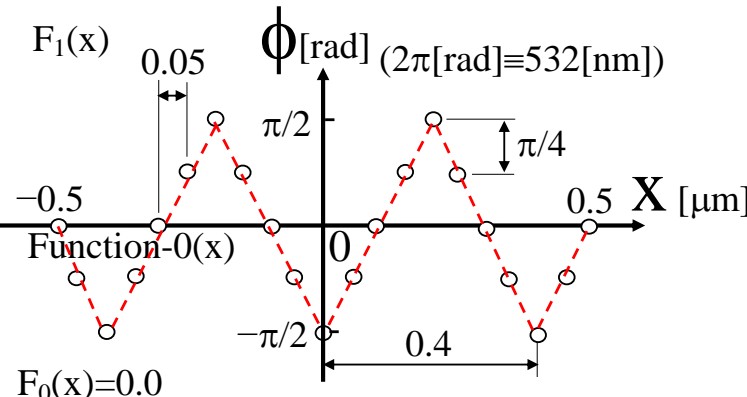

**Figure 14.** Phase distribution simulated for a periodic structure.

In this case, the linearity of the measurement results based on the results shown in Figures 9 and 12 was also considered, and a shape composed of a relatively small phase change was observed.

The results are shown in Figure 15. In this case, the position of 0 on the *x*-axis corresponds to the position of the optical axis. The phase distribution at 90 μm, which is the standard information, is shown in Figure 15a for when the phase of all light sources was set to 0 rad, in the same way as in the experiments using two and three light sources. In this case, the phase distribution was also converted to length h by replacing $2\pi$ rad with a wavelength of 532 nm, and the value is shown in round brackets.

In this case, the detected sinusoidal bias component (indicated by the red line) was not necessarily 0, as shown in Figure 14, but was 0.01 rad because the phase distribution had 90 μm set as the detection position. It is not possible to precisely investigate whether this result was caused by setting the optical path length to 90 μm from the measurement object surface since the light passed through the lens.

Even for phase distributions that were more complicated than in the cases using two and three light sources, as shown in Figure 15c, the phase information was maintained within the zeroth-order diffracted light, and the shape of the original phase distribution could be retrieved after passing through the lens. In this case, the length at which a change was extracted was small. In addition, the period of the phase distribution shown in Figure 15c was slightly larger, 0.76 μm, while the period of the periodic structure shown in Figure 14 was 0.4 μm. Although the magnification was originally supposed to be 1.2× based on the positional relationship between the position of the measurement object and the imaging plane, the magnification was larger than that. The zero-order diffracted light showed a slight spread immediately after being emitted from the light source, and although the beam converged after passing through the lens, it still reached 90 μm in a spreading state. Because the beam was slightly wider than the object to be measured, it was assumed that the phase distribution (shape distribution) was wider than 0.4 μm. This phenomenon can be observed in the region around 90 μm in Figure 6a.

This problem can also be observed in the experimental results shown in Figure 7 for a diffraction grating with a period of 278 nm, for which the measurement result was detected with a magnification of approximately 300 nm. This phenomenon of magnification change has often been observed in previous experiments. If this phenomenon can be corrected by setting up a pinhole in the simulation and if information can be extracted only in the area near the optical axis, it is possible that the change in magnification can be improved by collecting information at a position closer to the image formation position. In the future, it will be necessary to develop a super-resolution technology using the phase distribution by re-examining the simulation model using pinholes and scattered light.

However, with regard to observing microstructures, situations exist where an objective lens must be used to magnify the image. In this case, only information near the optical axis of the lens needs to be extracted using a pinhole. Figure 6 shows how the magnification

changes could be improved by collecting the information closer to the image formation position as opposed to the current 90 μm detection position. In the future, it will be necessary to investigate in detail the situation in which measurement sensitivity and magnification change with respect to the detection position.

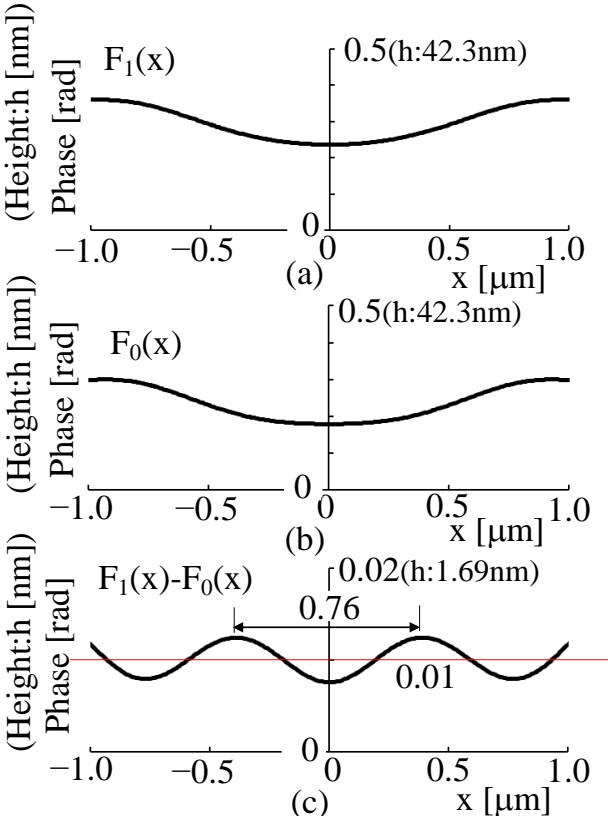

**Figure 15.** Results of the measurement model. (**a**) Standard phase distribution when phase value of all points was set as 0 rad. (**b**) Results including standard phase distribution. (**c**) Results with standard phase distribution eliminated.

In this study, by detecting the phase distribution of light using a simulation, it was clarified that the optical explanation for the realisation of super-resolution is the conservation of the phase distribution with respect to the shape of the measurement object in zero-order diffracted light.

## 4. Conclusions

The conventional concept of a diffraction limit associated with diffraction as proposed by Abbe and Rayleigh is based solely on the intensity distribution of light. In this situation, along with the diffracted light generated when the microstructures are illuminated, higher-order diffracted light, except for zero-order diffracted light, cannot pass through the lens aperture, according to Abbe's image formation theory. Therefore, it was believed that they could not be resolved in the imaging plane.

However, the phase distribution analysis performed in this study shows that zeroth-order diffracted light passes through the lens and that there is phase information in the zeroth-order diffracted light. As a result, it was confirmed in this study that the observation of microstructures beyond the diffraction limit can be realised by detecting the phase information contained in zeroth-order diffracted light with high resolution based on speckle interferometry.

In the future, a new simulation model that more closely imitates an actual optical system will be created, and a simulation using scattered light and extracting only zeroth-order diffracted light by utilizing pinholes will be performed.

**Funding:** This research was funded by JSPS KAKENHI, grant number 20H02165.

**Institutional Review Board Statement:** Not applicable.

**Informed Consent Statement:** Not applicable.

**Data Availability Statement:** Data are contained within the article.

**Acknowledgments:** I would like to thank Dahai Mi of Keisoku Engineering System Co. Ltd. for his kind guidance and support in the use of COMSOL Multiphysics.

**Conflicts of Interest:** The author declare no conflict of interest.

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
