# Peer review of "Role of Phase Information Propagation in the Realisation of Super-Resolution Based on Speckle Interferometry"

_photonics, doi:10.3390/photonics10121306_

Round 1

Reviewer 1 Report

Comments and Suggestions for Authors

The paper submitted by Yasuhiko entitled ‘Role of phase information propagation in the realisation of super-resolution based on speckle interferometry’ focus on the phase component of the zero-order diffracted light reflected from the microstructure. The mechanism for achieving super-resolution in Arai method was physically clarified. I would like to ask some questions and provide the comments to the authors.

1.      Briefly describe the principle of Arai method in the introduction, which is the foundation of the article.

2.      In Figure 2, the lens had a magnification of 200x and a numerical aperture of 0.62, which provided a diffraction limit of 660 nm. Why 0.62 NA not higher NA you used in this manuscript? The diffraction limit of 660 nm is too larger than traditional diffraction limit 250 nm. This method is suitable for imaging system with higher NA?

3.      What is the limitation of this super-resolution technology using speckle interferometry? Such as Structured illumination microscopy improves by 2-fold relative to widefield resolution.

4.      Is this super-resolution technology suitable for deep tissue imaging?

Comments on the Quality of English Language

However, the language is needs to be improved to make it easier for understand. 

Reviewer 2 Report

Comments and Suggestions for Authors

In this paper, the author investigated the physical reasons why it is possible to observe microstructures beyond the Rayleigh criterion by analyzing the phase distribution of light. The simulation results confirmed that the phase component of the zero-order diffracted light reflected from the microstructure and able to pass through the lens system contained the phase information related to the shape of the measured object. Overall, the idea in this paper is convincing while demonstrating another possibility of realizing super-resolution based on speckle interferometry through phase analysis. Although this article lacks real experiments, its innovativeness is sufficient for its acceptance on Photonics.

Comments on the Quality of English Language

There are some minor grammatical errors in the article. The authors should check and correct them carefully.

Reviewer 3 Report

Comments and Suggestions for Authors

The article proposes a practical approach to optical imaging of microstructures with super-Rayleigh resolution. The use of optical phase distributions to obtain super-resolution is a long-standing topic. Nevertheless, the author offers his own analyzing technique for speckle patterns, justifying it by a simulation of the phase distribution. Despite the usefulness of the results obtained, the presentation of the work requires some clarification and refinement in a number of places to help the reader's understanding. 

I recommend accepting the article for publication after minor revision. Below are my comments on the text.

1. Lines 21-25: Here it would be necessary to clearly distinguish between fluorescent methods and speckle interferometry, since these are different technologies for obtaining super-resolution.

2. Figure 2: In the view of the technique described in section 2.1, does this scheme include scanning in the transverse plane? Please clarify

3. Lines 141-142 "Va is the electromagnetic field intensity": probably "intensity" should be "amplitude". Judging by the complex exponential in equation (1), is "E" the electromagnetic field strength? It is worth clearly defining all the quantities in formula (1).

4. Caption to figure 3: it is necessary to indicate which physical quantity is being modeled. What is displayed on the 2D pictures, the intensity or strength of the electromagnetic field?

5. Caption to figure 4: The legend needs to be clarified. These pictures apparently show interference patterns in the diffraction and non-diffraction cases. It is necessary to specify the distribution of which parameter was built.

6. Line 282 "SEM": this abbreviation needs to be deciphered. Is it "scanning electron microscope"?

7. Caption to figure 7: The legend must be clarified. It is necessary to specify by what method each of the pictures (a) and (b) was obtained. "Measured result"? Is this measured by speckle interferometry? This needs explanation.

8. Line 416 "the phase at a position at 90 µm": Stylistically, something like "the phase at a distance of 90 µm" would be better here.

9. Line 491, "phase distribution as the object light". Maybe there should be "of" instead of "as"?

10. Caption to Figure 14. The legend needs to be reformulated. Maybe like this "Phase distribution simulated for a periodic structure"?

Reviewer 4 Report

Comments and Suggestions for Authors

The article is devoted to phase distribution analysis of light passing through a single biconvex lens at various planes between the source plane and the image plane. To do this author uses COMSOL simulation software to investigate light propagation in optical system attributed to speckle interferometry scheme. It is shown that microstructures beyond the diffraction limit can be observed by detecting the phase information contained in zeroth-order diffracted radiation.

Information is given in detail and is easy to understand.

Some minor issues:

1.       Line 48 – measured object?

2.       Fig.1. Figures (a) and (b) are too close. I would recommend to increase the distance. Arrow below x looks like a vector sign for f(x) in figure (b).

3.       Fig.2 would benefit from adding example of the intensity distribution registered by camera. All abbreviations should be explained either in the text or in figure caption.

4.       Pine hole is commonly referred to a diaphragm with a tiny hole. Such as that installed in collimator in the figure. The diaphragm shown between Lens and BS is not like that.

5.       Line 136. What is the essence of perfectly matched layer? Some explanations to the reader would be beneficial.

6.       The sentence between lines 123 and 127 is hard to understand. Please try to rephrase.

7.       Equation (1). “E” is not explained. Light intensity is commonly designated by “I”. Distribution set by equation (1) is not obvious. Can E(x,y) be also shown in Fig. 3 in 3D view? What are the dimensions of “E”? Is there a mistake in Eq(1)?

8.       Are all the linear dimensions measured in microns. Should not there be millimeters instead of microns? If this is true, then

8.1.    Line 185. 5.83 mm seems to be correct, not 5.83 um.

8.2.    Line 193. 70 mm seems to be correct, not 70 um.

8.3.    All units in Figs.5 and 6 should be changed from um to mm, except epsilon=2um and probably 1um.

8.4.    Line 317-372. 90um -> 90 mm. 50um -> 50 mm. 107.8um -> 107.8 mm

9.       If all linear dimensions of optical scheme are given and simulated in microns, will there be change in phase simulation results in case of real lens diameter of 30 millimeters? I worry about relation between epsilon=2um and lens diameter equal 30 microns and the second case of epsilon=2um and lens diameter equal 30 millimeters.

10.   Fig. 4. At what plane spatial distributions in (b) and (d) are shown? It is stated “At focus point”. According to Fig.3b this focal plane is placed at a distance f=37.8 from the lens. At the same time line 227 tells us that this is “the intensity distribution on the image plane”. According to Fig.3a image plane is 84.5 away from the lens. And according to Figs. 5 and 6 image plane is 154.5 away from the lens. Where those planes in (b) and (d) are located? Please clarify this.

11.   What are the units of the color scale bars in Figs. 5 and 6?

12.   Line 286. It is not clear what results are shown in Fig. 7b. Is this a real intensity distribution registered by camera in Fig.2 or some simulation results using configuration in Fig.3? More detailed description should be given here.

Article can be accepted after issue addressing.

Reviewer 5 Report

Comments and Suggestions for Authors

 This study demonstrates the  possibility of realising super-resolution based on speckle interferometry. The authors used electromagnetic field simulations. the paper present significance to the field of super-resolution and optical microscopy. However, it is not clear the novelty of the work when compared with reference 25 form same authors. I suggest a comparison and the authors to clarify more the novelty in the introduction part. The reference list should be updated, 22 percent are self citations.

Round 2

Reviewer 1 Report

Comments and Suggestions for Authors

Thank you very much for your thorough review of our manuscript.